# Older People Living in Nursing Homes: An Oral Health Screening Survey in Florence, Italy

**DOI:** 10.3390/ijerph16183492

**Published:** 2019-09-19

**Authors:** Fabrizio Chiesi, Maddalena Grazzini, Maddalena Innocenti, Barbara Giammarco, Enrico Simoncini, Giuseppe Garamella, Patrizio Zanobini, Caterina Perra, Lorenzo Baggiani, Chiara Lorini, Guglielmo Bonaccorsi

**Affiliations:** 1Medical Specialization School of Hygiene and Preventive Medicine, University of Florence, Viale GB Morgagni 48, 50134 Florence, Italy; 2Department of Health Sciences, University of Florence, Viale GB Morgagni 48, 50134 Florence, Italy; maddalena.grazzini@hotmail.it (M.G.); chiara.lorini@unifi.it (C.L.); guglielmo.bonaccorsi@unifi.it (G.B.); 3Medical Specialization School of Hygiene and Preventive Medicine, University of Florence, Viale GB Morgagni 48, 50134 Florence, Italy; maddalena.innocenti@unifi.it (M.I.); barbara.giammarco@unifi.it (B.G.); enrico.simoncini@unifi.it (E.S.); giuseppe.garamella@unifi.it (G.G.); patrizio.zanobini@unifi.it (P.Z.); 4Central Tuscany LHU, Piazza Santa Maria Nuova 1, 50134 Florence, Italy; caterina.perra@uslcentro.toscana.it (C.P.); lorenzo.baggiani@uslcentro.toscana.it (L.B.)

**Keywords:** oral health, dental care, elderly frailty, nursing home, cognitive state, functional autonomy, malnutrition risk

## Abstract

The oral health state plays an important role in the concept of ‘elderly frailty’, since institutionalized older people are prone to suffering from bad oral conditions. The aim of this study is to assess the state of oral health in the older residents of nursing homes and to measure its potential association with the cognitive state, the degree of functional autonomy, and the malnutrition risk. Methods: We enrolled 176 subjects from 292 residents in five nursing homes in Florence. For each subject, we performed the Malnutrition Universal Screening Tool, the Pfeiffer test, the Minimum Data Set—Long Form, a dental examination, and the Geriatric Oral Health Assessment Index questionnaire. The results show that the oral condition was poor in 43.8% of cases, medium in 38.1%, and good in 18.2%. A worse oral health state was significantly associated (*p* < 0.05) with a worse cognitive state and with a higher dependency in daily living activities. The malnutrition score among the older people was unrelated to the oral health condition (*p* = 0.128). It can be concluded that the oral health condition in older institutionalized subjects is an open challenge for the public healthcare system, since the maintenance of adequate good oral health is an essential element of good physical as well as cognitive and psychological health.

## 1. Introduction

With an expected population of 8.6 billion in 2030, 9.8 billion in 2050, and 11.2 billion in 2100 [1], and the proportion of older people rising, the healthcare needs for these people are increasing as well. Ageing is currently occurring throughout the world, both in the richest and the poorest nations: at present, there are an estimated 962 million people aged 60 or over (13% of the global population), projected to be 1.4 billion in 2030, 2.1 billion in 2050, and 3.1 billion in 2100 [1].

In Western countries, ageing represents a challenge for the sustainability of the social and healthcare systems. Europe has the greatest percentage of population aged 60 or over (25%) [1], and Italy represents one of the eldest countries: people aged 65 or more constitute 22.6% of the population [2], expected to rise up to 26.9% in 2030, reach a maximum of 33.9% in 2050, and then stabilize around 33.3% in 2065 [2]. In Tuscany, ageing is a hard question to face: older people represent 25.2% of all residents, and the same proportion is found in the territory of Florence and its surroundings, which is the area of our research [2].

This demographic change has introduced the need to address the concept of ‘elderly frailty’, which is a biological age-dependent state characterized by a reduced resistance to stress—secondary to the cumulative decline of several physiological systems—and an increased risk of negative health outcomes, such as functional decline, co-morbidity, disability, falls, institutionalization, and mortality [3,4,5]. Although the concept of elderly frailty has not been unanimously accepted yet, oral health condition certainly plays an important role in its net causation, creating a complex relationship of interdependence between oral and general health. In fact, the efficiency of the stomatognathic system not only affects the masticatory capacity with consequent nutritional effects but also determines social and family integration, being involved in activities like speaking, smiling, and expressing without pain and discomfort [6]. Globally, poor oral health among older people results in a high level of tooth loss, dental caries, a high prevalence of periodontal disease, xerostomia, and oral precancer/cancer; poor oral conditions represent a heavy burden, particularly for edentulous people [6]. Tooth loss can be disabling and can have a profound impact on the lives of some people [7]. Indeed, the World Health Organization (WHO) recognizes edentulism as a severe physical disability, being the cause of a set of clinical, functional, nutritional, psychological, and relational difficulties.

Oral health seems to be associated with both the cognitive state and the degree of functional autonomy in daily activities: a lower cognitive function is associated with a poor oral hygiene condition, tooth loss, and unstable dentures [8,9], while the number of remaining teeth has been found to be significantly associated with the level of physical activity in older people: people with 20 or more remaining teeth are more active than those with fewer than 20 [10]. Strömberg et al. show how the oral-health-related quality of life for older people who are on supportive care for daily living correlates strongly with the total number of teeth [11].

Among older people, those who are institutionalized are prone to suffering from worse oral conditions than those who live in their homes, and often have a greater difficulty in accessing oral health services for socio-economic reasons. Poor oral health favours the onset of preventable diseases and conditions that threaten the quality of life in general, and specifically for subjects who are dependent on others for care, which is very common among older people [12]. In particular, edentulism and bad oral health are associated with the risk of malnutrition, i.e., weight loss, a low BMI and sarcopenia [13,14]. On the other hand, some authors suggest that good oral health can play an important role in tackling malnutrition [15]. The relation between a low nutritional intake and bad oral health in the institutions can depend on both the specific diet needs of people who have suffered a loss of masticatory ability, as well as on their poor general condition (i.e., compromised cognitive state) and comorbidities that directly affect the nutritional state [16,17]. In nursing homes (NHs), which are often overcrowded owing to the increasing requests for admissions, diet and oral health represent two fundamental aspects of the quality of care provided, meaning that their measure cannot be ignored [18].

For these reasons, according to the literature, the objective of this research was to assess the oral conditions in the older residents of NHs by means of a multidimensional approach, which takes into account the cognitive state, the degree of functional autonomy, and the malnutrition risk, and to measure the potential association between oral health and these factors.

## 2. Materials and Methods

This is a cross-sectional study. The study was conducted in Italian.

We contacted all the 35 NHs, of which 31 are private and four are public, to recruit the subjects located in the Florence Health District, Italy. All the structures are authorized and accredited by the Tuscany Region and offer the same services, at the same costs. The differences in the tariffs paid by the people are due to the co-participation of the regional government (i.e., co-payment), assessed in terms of healthcare as well as social degree of dependency. Five nursing homes (14.3%)—two private and three public—voluntarily participated in the project. The participating NHs hosted a total of 292 residents, which is to say 20% of the total older persons who lived in the 35 NHs at the end of 2017. In all, 176 residents (60%) participated, and the adherence varied in the five structures from a minimum of 43% to a maximum of 70%.

An informative letter describing the objectives and explaining the tests and the ways of administration was shared with each participating institution, together with a sheet for collecting information about practices and measures currently adopted in the structures (i.e., screening tools for malnutrition risk, methods of recording the body weight, and nursing assistance during meals). Within 60 days, after collecting all the documents, we shared the days for administering the chosen tests with each NH Director.

We enrolled only those subjects who voluntarily joined the project and signed the informed consent form, which was provided directly by the subject or by his/her legal representative in cases of severe cognitive impairment. We excluded all the people who did not give consent, or whose clinical conditions compromised their participation. For each enrolled subject, we collected the demographic as well as anthropometric data—i.e., name, surname, birth date, gender, education, current weight, height, and the weight recorded until 3–6 months before; all these data were recovered from the medical records of the residents.

For each subject, the nutritional risk assessment was performed using the Malnutrition Universal Screening Tool (MUST). MUST attributes a score to the BMI (>20 = 0 points, 18.5–20 = 1 point, <18.5 = 2 points), to unexplained weight loss (<5% = 0 points, 5–10% = 1 point, >10% = 2 points), and to acute illness effects (2 points). The sum of these three components classifies subjects into three malnutrition risk classes (0 = low risk, 1 = medium risk, and ≥2 = high risk) [19].

The cognitive state was assessed by the short portable mental state questionnaire (known also as the Pfeiffer test), composed of 10 questions. The test, which also includes a correction factor for education (−1 if the subject only finished primary school, +1 if the subject attended university), assigns a final score based on the number of errors, and allocates the respondents into four categories (score: from −1 to 2 = undamaged intellectual function, 3−4 = low cognitive impairment; 5–7 = medium cognitive impairment, and 8−11 = severe cognitive impairment) [20].

The Dependency in Activities of Daily Living (ADLs) was evaluated by the items of the functional status included in the Minimum Data Set (MDS)—Long Form (MDS-ADL), which attributes a score from 0 (complete independence) to 4 (need for total assistance) to seven ADLs. The final score divides subjects into four classes: 0–7 points = complete independency, 8–14 points = low dependency, 15–21 points = medium dependency, and 22–28 points = high dependency [21].

For each subject, the dentist of the research group, who is skilled in the orthogeriatric approach, has performed an assessment of oral health conditions using a dental visit checklist composed of six items (dental visit numbers in the last two years, the reason of the last dental visit, the oral mucosa condition, edentulism, number and an assessment of dentures) with a final judgment (poor, medium, or good oral health condition). Patients who were totally edentulous without dentures were assigned to the poor level. Patients that were totally edentulous with dentures who presented at least two of the problems from lack of support, stability, retention, incorrect border extensions and incorrect palatal seal were assigned to the poor level. Patients who were partially edentulous with fewer than three contacts between opposing teeth, natural or artificial, were assigned to the poor level. Patients with denture-induced stomatitis and/or inflammatory papillary hyperplasia and/or chronic atrophic candidiasis were assigned to the poor level. Patients with debris/tartar/plaque in most areas of the mouth or on most of the denture or severe halitosis were included in the poor level. Patients who were totally edentulous with dentures that presented a good support, stability, retention, correct border extension, and correct posterior palatal seal were assigned to the good level. Patients who were partially edentulous with more than 20 contacts between opposing teeth, natural or artificial, were assigned to the good level. All other patients were assigned to the medium level.

In addition, each subject with an eligible cognitive state (Pfeiffer Test score <5) answered the Geriatric Oral Health Assessment Index (GOHAI) questionnaire to evaluate the subjective perception of one’s oral health condition. The questionnaire comprised 12 questions and a final classification into three levels (poor, medium, or good perception of the oral health condition) [22]. We used the Italian version of the GOHAI, which has already been assessed by Franchignoni et al. [23].

Teeth loss in older people is considered an important factor in assessing the healthcare condition of the oral cavity [24]. Obviously, the presence or absence of opposing teeth, their periodontal status, and the absence of decay are other important variables. Based on recent literature [25], we also used this operational three-level classification, considering minimal edentulism (0–12 missing teeth), partial edentulism (13–31 missing teeth), and complete edentulism (32 missing teeth). From the literature, we retrieved other validated instruments to evaluate the oral assessment, like the Revised Oral Assessment Guide (ROAG) and the Oral Health Assessment Tool (OHAT). In this study, our main interests were focused on some items that are not in the ROAG instrument, such as the functionality of the masticatory apparatus and the evaluation of the fit of the dentures [26,27].

All the measurements were carried out by a multidisciplinary team, composed of a dentist and several doctors in specialist training at the School of Hygiene and Preventive Medicine of the University of Florence, with the support of the staff of the NHs. The dental visit was performed with the use of a mirror, dental probe, no magnifying lens, and during one single appointment under natural light. Each specialist filled the data sheet referring to his specialization; thus, only one examiner filled all the data for the same index. No reliability check was performed. All the residents were assessed individually. The team collected personal data on a paper sheet and assigned a progressive identification number (ID) to each subject in order to guarantee anonymity. All the demographic, dental, and medical data were recorded in an electronic database for processing.

The data processing and analysis were conducted using IBM SPSS Statistic 25^tm^ (SPSS, Inc., Chicago, IL, USA). A descriptive analysis was performed, and the data were presented as a percentage, mean ± standard deviation (SD), or median + range (Min.; Max.). Associations were tested using the Chi-Square test for categorical variables and the Kruskal-Wallis test for continuous variables. For each test, we considered a significance level *α* = 0.05.

The study was developed according to a Regional Decree (DGRT 426/2014 implementation of dental services in Tuscany). The project was approved by the Ethics Committee of Area Vasta Centro on 27 April 2017 (Rif. CEAVC: OSS.16.222). The study was conducted according to the principles of the Helsinki Declaration.

## 3. Results

The examinations of the residents started in October 2017 and were completed by October 2018.

The descriptive analysis of all the detected variables is shown in Table 1. The mean age was 84.4 years (± 8.3 SD)—67.6% were women and 32.4% men. The majority of subjects (71%) did not have any educational qualification or at most had finished primary school, 11.9% had graduated from junior high school, 13.6% had a high school graduation, and 3.4% had a degree.

The dental examination revealed that only 21% of subjects had minimal edentulism (0–12 missing teeth); 39.2% had partial edentulism, and the remaining 39.8% had complete edentulism. The overall clinical assessment of the oral health condition performed by the dentist resulted as poor in 43.8% of cases, medium in 38.1%, and good in 18.2%. In contrast, the subjective evaluation of one’s own oral cavity wellbeing via GOHAI resulted in ‘good’ for almost the whole sample (98.9%), revealing a sound contrast compared to the dentist’s assessment.

The Pfeiffer test shows that 34.7% of subjects had an undamaged intellectual function, 17.6% and 17% had low and medium functions, respectively, and 30.7% suffered from severe cognitive impairment. The assessment of daily activities indicated that 22.2% were completely independent, 21% had a low dependence, 21% had a medium dependence, and 35.8% had a high dependence level.

The mean BMI was 23.77 kg/m^2^ (± 5.18 SD) for a total of 162 residents because of a lack of data (weight or height) for 14 subjects. Almost half of the cases (48.1%) resulted in the range of normal weight for height (BMI range of 18.5 to 25 kg/m^2^), 14.2% were underweight (BMI < 18.5 kg/m^2^), 25.9% were overweight (BMI range of 25 to 30 kg/m^2^), and 11.7% were obese (BMI > 30 kg/m^2^).

The MUST could only be computed for 160 residents, since the assessment of the weight in the previous 3–6 months was not reported in the medical records of two subjects. In the remaining sample, 70.6% had a low malnutrition risk, 11.9% had a medium malnutrition risk, and 17.5% had a severe malnutrition risk.

The potential associations between the general dental assessment and other healthcare conditions were investigated (Table 2 and Table 3). The age distribution does not present significant differences in the three categories of the overall dental assessment (*p* = 0.694). Moreover, the dentist’s assessment seems to be associated neither with gender (*p* = 0.538) nor education (*p* = 0.806).

The association between the oral health condition and the Pfeiffer test scores shows that the distribution of the scores (ranging from −1 to 11) of the Pfeiffer test is significantly associated (*p* < 0.05) with the oral health condition: the more preserved the cognitive functions are, the better the oral health condition is.

Furthermore, the degree of daily living autonomy, as assessed by MDS-ADL, seems to influence the oral health condition; in fact, a statistical analysis shows a significant association (*p* < 0.05) between the general oral cavity assessment and MDS-ADL assessment, considering either its score distribution or its classification into four categories. Specifically, subjects with a high dependency level in daily living activities have a greater risk of a compromised oral health.

Our study shows that the malnutrition score among the older people, measured by MUST, is not associated with the oral health condition: in fact, the general dental assessment does not seem to affect malnutrition (*p* = 0.128).

## 4. Discussion

Our results suggest that almost 80% of older residents in NHs have poor to medium oral health conditions, confirming the data already found in the literature [28]. The main role in determining this poor oral condition is played by edentulism—79% of the residents included in this study have partial or complete edentulism, and this seems, as suggested by literature, to have a strong impact on their quality of life [7,29].

It is worth mentioning that, in our sample, almost all the older subjects perceive their own oral health condition as being good: this subjective perception contrasts with the dentist’s overall assessment. Despite the undamaged cognitive state, our hypothesis is that institutionalized older people get used to their poor oral condition over time and develop compensatory behavioural strategies. This behaviour risks being even more dangerous for the residents’ oral health because it could lead to even more subjective negligence: consequently, it is a fundamental and specific responsibility of NH professionals to pay attention to the oral hygiene of older people as an indicator of the quality of care.

The present study suggests a positive and significant association between the oral health condition and preserved cognitive functions, as already described by other authors [30,31]: cognitive decline may affect the nutritional intake and musculoskeletal oral coordination, worsening the ability to provide efficient oral hygiene and to access dental care [13]. Moreover, a good oral condition enables social and family integration, lowering the risk of isolation and depression. To confirm these assumptions, it is necessary to recruit research that adopts both larger samples and a different study design such as a perspective cohort study.

The dependency level in daily living activities was found to be significantly associated with better oral conditions: these results are confirmed by other studies in the literature [32], since institutionalized subjects, who cannot autonomously perform daily living activities and thus are not able to take care of their own oral health, are more likely to suffer from dental disorders such as edentulism and mucositis. Functional impairment, in combination with dependency, can imply an increased risk of dental disease and tooth loss, as well as a poorer ability of these individuals to manage their dental hygiene and tolerate dental treatment [33].

Malnutrition among institutionalized older people is a widespread and underestimated question—with important repercussions in terms of morbidity, mortality, increase in healthcare costs, reduced autonomy, and worse outcomes [34]. The risk of malnutrition in the enrolled NHs is consistent with the literature [34,35,36]. However, our study does not report that poor oral cavity conditions are significantly associated with a higher malnutrition score. This suggests that other factors could influence the nutritional state of these subjects. As suggested by other authors [37,38], apart from non-modifiable factors like age and comorbidity, other factors can be linked to modifiable determinants, like unappetizing food, a monotonous menu, few opportunities for conviviality, inadequate artificial feeding, overly restrictive diets, and a failure to monitor the weight and food intake. It is therefore necessary to introduce and apply some dietary concerns for institutionalized older people, in particular those with a cognitive deficit [39], in order to routinely screen every person for malnutrition, recognize people at risk of malnutrition at an early stage, and schedule nutritional interventions. Since weight loss is the most important sign of malnutrition, the weight should be monitored and recorded regularly. When these actions are put into practice, the prevalence or risk of malnutrition in NHs is lower [36].

Some authors draw attention to the gap in the research and development of effective strategies for improving the quality of oral healthcare in NH residents [40,41]. Nurses can play a critical role in improving oral healthcare for older adults [42]: the staff members of NHs often perceive residents to be in good oral health, even when the dental treatment needs have been ascertained by dentists [43]. Furthermore, our study highlights the need for training programs for NH professionals, as a means of introducing a routine surveillance of all the residents’ oral health condition and of facilitating, when necessary, the intervention of the dentist in preventing the exacerbation of the oral functions and in treating decays.

Educational interventions can improve nurses’ knowledge and accuracy of oral health assessment and care [42]. Oral-health-related programs can be improved by using a combination of interventions that engage everyone involved in the process of decision-making: specific training; surveillance to provide both quantitative and qualitative evidence; the adoption of good practices and appropriate guidelines; and new knowledge translation, based on the field data, to promote oral healthcare for older people in NHs [44,45].

Portella et al. describe how an oral hygiene educational program for caregivers, such as a two-hour lecture covering both the theory and the practice of oral hygiene, can have a positive impact on oral health conditions in institutionalized older people [46]. To improve oral health conditions in older people in NHs, Pronych et al. suggest the introduction and development of the role of an oral health coordinator, who can act as a liaison between the nursing and dental staff, providing resources for nursing assistants and ensuring residents’ daily oral care [47].

Additionally, executive directors in NHs understand the importance of oral health from the perspective of general health, so that one can propose strategies to increase their awareness and knowledge, so as to support the inter-professional practice of nurses working with dental staff in optimizing the oral healthcare of residents [48].

Whenever critical oral health conditions were found, we offered a dental treatment plan free of charge to each resident, in agreement with the resident’s general practitioner in charge of his assistance, to guarantee appropriate continuity of dental care, as well as providing all the participants, NH staff, and caregivers with an educational brochure containing advice on maintaining good oral and nutritional health.

The limitations of this study are primarily related to the mode of NH enrolment, which, being voluntary, might have produced a best-case scenario artefact, while limiting the generalisability of the results. NHs are quite closed and reserved communities, and only a small percentage (less than 15%) agreed to participate in the study. The five participating NHs probably represent the more concerned institutions.

Furthermore, the recruitment of people in each NH represented a limitation: we found difficulties in getting consent, especially for residents with a legal representative, who were often difficult to contact—at last, the final sample was formed of subjects who could provide consent autonomously or whose legal representative was easy to reach. In future, an information campaign to raise the awareness of residents and their legal representatives about the importance of the quality of primary dental care could and should be necessary as a means of increasing participation.

Finally, we believe that oral indicators should be part of the process of accreditation and a benchmark in the quality of care in NHs. In fact, Benedetti et al. highlight how a lack of relevant policies with a public health focus, an absence of systematic oral health surveillance, and a limited access to care for large population groups are strong indicators that oral health is not a political priority [49]. For these reasons, it is necessary—perhaps more so than in the past—to develop tools to facilitate analyses, discussions, and change in order to improve the political priority of oral health in Italy.

## 5. Conclusions

Having dentures not only represents the possibility of feeding oneself or of rediscovering the joy of smiling and having social relations without stigma; it is also assures dignity to every single subject, wherever he/she lives. The maintenance of an appropriately good oral health is an essential element of good physical as well as cognitive and psychological health.

The relationship between oral health and good nutrition, whether confirmed or not, appears sometimes to have a lower importance in terms of quality of life when compared to other factors, such as self-esteem, integration in the community, and the capacity to speak and smile without stigma.

The oral health condition among older institutionalized subjects is an open challenge for the public healthcare system. More studies are necessary to better define the tools, interventions, and professionals that should be involved in providing the appropriate services needed by these people.

## Figures and Tables

**Table 1 ijerph-16-03492-t001:** The descriptive analysis of variables in a study of oral health in nursing homes in Florence, Italy, October 2017–October 2018 (*n* = 176).

Variables	Mean (SD)
**Age**	84.4 years (± 8.3)
**BMI**	23.77 (± 5.18)
**Variables**	**Frequencies (Percentage)**
**Gender**	
Female	119 (67.6%)
Male	57 (32.4%)
**Education**	
None/Primary school	125 (71%)
Junior high school	21 (11.9%)
High school	24 (13.6%)
University	6 (3.4%)
**Edentulism**	
0–12 missing teeth	37 (21%)
13–31 missing teeth	69 (39.2%)
32 missing teeth	70 (39.8%)
**Overall clinical assessment of oral health condition**	
Poor	77 (43.8%)
Medium	67 (38.1%)
Good	32 (18.2%)
**GOHAI (*n* = 92) ***	
Poor	0 (0%)
Medium	1 (1.1%)
Good	91 (98.9%)
**Pfeiffer Test**	
Undamaged intellectual function	61 (34.7%)
Low cognitive impairment	31 (17.6%)
Medium cognitive impairment	30 (17%)
High cognitive impairment	54 (30.7%)
**MDS-ADL**	
Complete independency	39 (22.2%)
Low dependency	37 (21%)
Medium dependency	37 (21%)
High dependency	63 (35.8%)
**BMI categories (*n* = 162) ****	
Underweight	23 (14.2%)
Normal	78 (48.1%)
Overweight	42 (25.9%)
Obese	9 (11.7%)
**MUST (*n* = 160) ****	
Low malnutrition risk	113 (70.6%)
Medium malnutrition risk	19 (11.9%)
High malnutrition risk	28 (17.5%)

Where the bold is present, it is necessary. SD, Standard Deviation; BMI, Body Mass Index; GOHAI, Geriatric Oral Health Assessment; MDS-ADL, Minimum Data Set of Activities of Daily Living; MUST, Malnutrition Universal Screening Tool. * GOHAI was not administrable for 84 subjects (Pfeiffer test >4 errors). ** percentages referred to those subjects for whom medical records were available and complete.

**Table 2 ijerph-16-03492-t002:** The distribution of the enrolled subjects for the oral health general assessment and the other collected data in a study of oral health in nursing homes in Florence, Italy, October 2017–October 2018 (*n* = 176).

	Oral Health General Assessment			
Variables	Poor	Medium	Good	*p* Value *
**Gender**				0.538
Female	53 (44.5%)	47 (39.5%)	19 (16.0%)	
Male	24 (42.1%)	20 (35.1%)	13 (22.8%)	
**Education**				0.806
None/Primary school	55 (44%)	48 (38.4%)	22 (17.6%)	
Junior high school	11 (52.4%)	8 (38.1%)	2 (9.5%)	
High school	9 (37.5%)	9 (37.5%)	6 (25%)	
University	2 (33.3%)	2 (33.3%)	2 (33.3%)	
**Pfeiffer Test (categories)**				0.247
Undamaged intellectual function	21 (34.4%)	23 (37.7%)	17 (27.9%)	
Low cognitive impairment	13 (41.9%)	13 (41.9%)	5 (16.1%)	
Medium cognitive impairment	14 (46.7%)	12 (40.0%)	4 (13.3%)	
High cognitive impairment	29 (53.7%)	19 (35.2%)	6 (11.1%)	
**MDS-ADL (categories)**				0.038
Complete independency	14 (35.9%)	14 (35.9%)	11 (28.2%)	
Low dependency	11 (29.7%)	15 (40.5%)	11 (29.7%)	
Medium dependency	17 (45.9%)	16 (43.2%)	4 (10.8%)	
High dependency	35 (55.6%)	22 (34.9%)	6 (9.5%)	
**MUST**				
Low malnutrition risk	46 (40.7%)	44 (38.9%)	23 (20.4%)	
Medium malnutrition risk	13 (68.4%)	6 (31.6%)	0 (0%)	0.128
High malnutrition risk	13 (46.4%)	11 (39.3%)	4 (14.3%)	

Where the bold is present, it is necessary. MDS-ADL, Minimum Data Set of Activities of Daily Living; MUST, Malnutrition Universal Screening Tool. * Chi-Square test.

**Table 3 ijerph-16-03492-t003:** The age, score at the Pfeiffer test and score at MDS-ADL for the oral health general assessment in a study of oral health in nursing homes in Florence, Italy, October 2017–October 2018 (*n* = 176).

Oral Health General Assessment				
Median (Min;Max)	Poor	Medium	Good	*p* Value *
**Age (years)**	87 (54;104)	86 (56;99)	85.5 (66;98)	0.694
**Pfeiffer Test score**	5 (−1;11)	4 (−1;10)	2 (−1;11)	0.048
**MDS-ADL score**	19 (0;28)	17 (0;28)	12 (0;28)	0.019

Where the bold is present, it is necessary. MDS-ADL, Minimum Data Set of Activities of Daily Living; * Kruskal–Wallis Test.

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
