# Peer review of "Older People Living in Nursing Homes: An Oral Health Screening Survey in Florence, Italy"

_ijerph, 2019, doi:10.3390/ijerph16183492_

Round 1
Reviewer 1 Report
English language: There needs to be some editing in the use of English. There is overuse of colons and semicolons. The text includes run-on sentences. There are also some problems with sentence structure, word choice, and capitalization.
The major problem with this paper is the definition of oral health used. Authors discuss edentulism to represent the most serious level of poor oral health. This is not accurate. If an individual has well fitting dentures, can eat properly, and has no pain they would be considered by dental professionals to have good oral health. Likewise the categorization of participants into three groups based on number of teeth retained is problematic. In addition to number of retained teeth, oral health depends upon the position of the teeth in the mouth and whether or not their is an opposing tooth. The three groups created are too broad and also inaccurate. The reference cited incorporates tooth positioning, not just number of teeth. Finally, the most serious oral health problems in an aging population are not mentioned at all. These are dental caries and periodontal disease.
The discussion and conclusions do not reflect the research results.
Author Response
English language: There needs to be some editing in the use of English. There is overuse of colons and semicolons. The text includes run-on sentences. There are also some problems with sentence structure, word choice, and capitalization.
The manuscript has been extensively revised by a professional proofreader.
The major problem with this paper is the definition of oral health used. Authors discuss edentulism to represent the most serious level of poor oral health. This is not accurate. If an individual has well fitting dentures, can eat properly, and has no pain they would be considered by dental professionals to have good oral health. Likewise the categorization of participants into three groups based on number of teeth retained is problematic. In addition to number of retained teeth, oral health depends upon the position of the teeth in the mouth and whether or not their is an opposing tooth. The three groups created are too broad and also inaccurate. The reference cited incorporates tooth positioning, not just number of teeth. Finally, the most serious oral health problems in an aging population are not mentioned at all. These are dental caries and periodontal disease.
We agree with the reviewer, but the scope of the article was to enlighten an aspect which has never been explored before in our territory. At this proposal, we added more considerations and references about oral health definition and the different levels used. Moreover, we specified that the use of the number of teeth retained is only one of the variables analysed for the assessment of oral health State.
The discussion and conclusions do not reflect the research results.
We improved the discussion adding new references, to reinforce our results.
Reviewer 2 Report
Thank you for interesting reading. The findings are not new but it's important that you look upon this problems from a multidisciplinary perspective and in that way highlight the issue for other professions. I's also always good to have reports from different countries.
Hovewer, there are some queries to handle before the manuscript eventually could be actual for publication.
The language has to be reviewed, there are some unlucky choise of words/expressions, some misspellings and some layout mistakes.
When it comes to the method, there need to be a better explanation how the nursing homes were recruited - announcements? Some selection before they volontary joined? The number of residents is described in Table 1 but not if they are private, catholic...? or focused on certain types of residents. Is the staff manning similiar, do the residents pay anything...? There is also missing information about how dentistry for elderly people in the area is arranged and how nursing staff provide oral care to the residents.
The dental checklist is probably a good choise in the context but there woukld be some words about why you noy use an established instrument like Revised oral assessment guide, ROAG, that also could have given some references to compare with.
Pfeiffers test is well known but not as a diagnostic instrument. It is most often used for screenings and this must be discussed.
When it comes to statistics, you report means and SD but use nonparametric methods for analyzes. I would prefer to report medians and ranges - or both.
The level "excellent" that is used both for the dental checklist and GOHAI seems unrealistic. The instruments are superficial and words like "acceptable or good" would be chosen.
Some references are old and I suggest that you update the list. Some suggestions are:
Lantto A, Lundqvist R, Wårdh I. Oral Status and Prosthetic Treatment Needs in Functionally Impaired and Elderly Individuals. Int J Prosthodont. 2018 Sept/Oct;31(5):494-501. Strömberg E, Holmén A, Hagman-Gustafsson ML, Gabre P, Wårdh I. Oral health related quality of life in homebound elderly dependent on moderate and substantial supportive care for daily living. Acta Odontol Scand. 2013 May-Jul;71(3-4):771-7. Ástvaldsdóttir Á, Boström A-M, Davidson T, et al. Oral health and dental care of older persons—a systematic map of systematic reviews. Gerodontology. 2018;00:1–15. Fereshtehnejad SM, Garcia-Ptacek S, Religa D, Holmer J, Buhlin K,Eriksdotter M,Sandborgh-Englund G. Dental care utilization in patients with different types of dementia:A longitudinal nationwide study of 58,037 individuals. Alzheimers Dement. 2018 Jan;14(1):10-19.
Author Response
The language has to be reviewed, there are some unlucky choise of words/expressions, some misspellings and some layout mistakes.
The manuscript has been extensively revised by a professional proofreader.
When it comes to the method, there need to be a better explanation how the nursing homes were recruited - announcements? Some selection before they volontary joined? The number of residents is described in Table 1 but not if they are private, catholic...? or focused on certain types of residents. Is the staff manning similiar, do the residents pay anything...? There is also missing information about how dentistry for elderly people in the area is arranged and how nursing staff provide oral care to the residents.
We thank the reviewer for these observation. We described more in detail the mode of recruitment and the organizing features of the nursing homes.
The dental checklist is probably a good choise in the context but there woukld be some words about why you noy use an established instrument like Revised oral assessment guide, ROAG, that also could have given some references to compare with.
We added discussion about the ROAG into the text and the references.
Pfeiffers test is well known but not as a diagnostic instrument. It is most often used for screenings and this must be discussed.
We do not use Pfeiffer test as a diagnostic tool (that is, whether the patient is affected by dementia or not), but only to assess the cognitive state. Pfeiffer test is the tool routinely used in all the public health structures in Tuscany since the first assessment before entering the NH.
When it comes to statistics, you report means and SD but use nonparametric methods for analyzes. I would prefer to report medians and ranges - or both.
As suggested by the reviewer, we have reported medians and ranges.
The level "excellent" that is used both for the dental checklist and GOHAI seems unrealistic. The instruments are superficial and words like "acceptable or good" would be chosen.
For dental checklist and GOHAI, we changed the word "excellent" with "good".
Some references are old and I suggest that you update the list.
We updated the references adding new and more recent references.
Reviewer 3 Report
This is a brief report of the oral health status and other characteristics of a nursing home population in Florence, Italy. The paper is mistitled. I suggest retitling it “Elderly People Living in Nursing Homes: An Oral Health Screening Survey in Florence, Italy.”
There is a sparse literature on the oral health of older adults from Italy so a report, however limited in scope, is welcome. It might be appropriate then to cite the paper by Benedetti and others (2015) on the priority of oral health in Italy.
There are several mandatory changes that will improve this manuscript:
Throughout, the manuscript the authors should clarify if these “nursing homes” are “case di repose”—use the accurate terminology for Italy. One would like to know if these are state run or private. Page 3, line 100. The authors should explain how many nursing homes were approached, how they were approached, and the reasons given for non-participation. We do not know how many nursing homes there are in Florence so it is impossible to judge how much one can generalize from these data? This information should be provided along with information about how typical the participating homes were. The apparent low participation rate contradicts the reference and statement on page 8, line 268 that the executive directors of nursing homes perceive the importance of oral health. The statement in line 101 is confusing. All of the information on the participation rate should be in one page (not also on page 4). Since the rate is quite low, the authors should provide greater detail about the reasons for non-participation and information on how these individuals might be different from other residents (age, sex, etc.). Table 1 is not necessary. On page 3, the authors need to clarify that the study was conducted in Italian language. They also need to provide appropriate citations to the Italian versions of these instruments. In the case of the Pfeiffer test and GOHAI, the validity of the Italian versions may never have been clearly established. See, for example, the brief paper by Franchignoni in 2010 on the GOHAI. Additional detail is also need on the actual procedure, setting, etc of the assessment. Also on page 3, line 132, the authors need to more clearly explain the criteria for poor, medium, and excellent oral health and provide information on the training of the examiner and on his/her reliability. The actual data on the parts of this apparent index need to be presented. The examination procedure should be described (for example, lighting) Likely, the authors cannot fully address the methodological concerns in this review, thus they will need to expand the limitations section on page 8. The English grammar needs review (see for example, the last paragraph (page 8). There is no reason to capitalize Nursing Homes throughout the paper. The link for reference 2 works but it does not lead specifically to the information being cited. When citing online sources, the authors should consider using WebCite or similar free services for archiving the source information. The journal name in ref 22, 23 is not properly capitalized. Overall, a spot check suggests the citations are accurate but all should be rechecked.
Author Response
The paper is mistitled.
We changed the title as suggested by the reviewer.
There is a sparse literature on the oral health of older adults from Italy so a report, however limited in scope, is welcome. It might be appropriate then to cite the paper by Benedetti and others (2015) on the priority of oral health in Italy.
We focused on the lack of policies on oral health in Italy, citing the paper by Benedetti and others (2015).
Throughout, the manuscript the authors should clarify if these “nursing homes” are “case di repose”—use the accurate terminology for Italy. One would like to know if these are state run or private. Page 3, line 100. The authors should explain how many nursing homes were approached, how they were approached, and the reasons given for non-participation. We do not know how many nursing homes there are in Florence so it is impossible to judge how much one can generalize from these data? This information should be provided along with information about how typical the participating homes were
As you suggested, we clarified the mode of recruitment and the organizing features of the nursing homes.
The apparent low participation rate contradicts the reference and statement on page 8, line 268 that the executive directors of nursing homes perceive the importance of oral health.
From previous interviews with the executive directors of nursing homes, they referred to perceive the importance of oral health, in line with reported by other authors (Wintch et al. 2014). At the same time, since the NHs are quite closed communities, it is hard to convince them to take part in scientific studies. This is the reason why we suggest, in the discussion, the need to develop easier tools to facilitate analysis, discussion and change in order to improve political priority of oral health in Italy.
The statement in line 101 is confusing. All of the information on the participation rate should be in one page (not also on page 4). Since the rate is quite low, the authors should provide greater detail about the reasons for non-participation and information on how these individuals might be different from other residents (age, sex, etc.). Table 1 is not necessary. On page 3, the authors need to clarify that the study was conducted in Italian language.
In according to the reviewer’s observation, we better specified the materials and methods.
In the case of the Pfeiffer test and GOHAI, the validity of the Italian versions may never have been clearly established.
There is no validated Italian version of Pfeiffer test, but this test is an universal tool used in public health to assess the cognitive state of patients and in our territory it is used everywhere both at the first visit (before entering the NHs), and in the subsequent assessments. We clarified the validity of the Italian version of GOHAI adding the specific reference (Franchignoni et al. 2010).
Additional detail is also need on the actual procedure, setting, etc of the assessment. Also on page 3, line 132, the authors need to more clearly explain the criteria for poor, medium, and excellent oral health and provide information on the training of the examiner and on his/her reliability. The actual data on the parts of this apparent index need to be presented. The examination procedure should be described (for example, lighting) Likely, the authors cannot fully address the methodological concerns in this review, thus they will need to expand the limitations section on page 8.
We have better described the three levels of oral health and the definition of Oral Heath used in the paper. We hope to have fixed these points into the new version of the text. About the examiner, we have better specified that the same professionals proceeded to all the assessments and all the dental assessment have bee made by the same dentist, so that it is not necessary to evaluate inter reliability. We have explained the examination procedure more in detail.
The English grammar needs review.
The manuscript has been extensively revised by a professional proofreader.
There is no reason to capitalize Nursing Homes throughout the paper.
We have corrected it.
The link for reference 2 works but it does not lead specifically to the information being cited. When citing online sources, the authors should consider using WebCite or similar free services for archiving the source information.
About the reference 2, the link is cited according to the indications provided by the journal. Data shown in the tables are referred to an url address that does not change during the navigation.
The journal name in ref 22, 23 is not properly capitalized.
We corrected it.
Overall, a spot check suggests the citations are accurate but all should be rechecked.
We rechecked and added new and more recent references.
Round 2
Reviewer 1 Report
none
Author Response
Thank you very much for your approval.
Reviewer 2 Report
Dear authors
Thank you for having improved the manuscript that I now consider suitable for publication.
Just one comment (that I honestly missed in my first reading): It's now more proper to use the expression "older" than "elderly". I suggest that you discuss with the editor if you should change the sentences where you have used "elderly" or if the text could be as it is in it's present form.
Author Response
Just one comment (that I honestly missed in my first reading): It's now more proper to use the expression "older" than "elderly". I suggest that you discuss with the editor if you should change the sentences where you have used "elderly" or if the text could be as it is in it's present form.
As suggested by the reviewer, we changed "elderly" with "older". According with the cited literature, we do not change the expression "elderly frailty".
Reviewer 3 Report
The authors have been responsive to recommended changes. There are, however, more changes needed.
Page 1, line 37. Substitute dental examination for dental visit.
Page 1, line 40. Rewrite the sentence beginning “The risk of malnutrition..” to read “The malnutrition score was unrelated to ….” An actual odds ratio was not actually calculated. Add a statistic to this sentence.
Page 3, line 205. Rewrite this paragraph to read “The objective of this research was….”
Page 3, line 211. Add, Florence, Italy
Page 3, line 215. The sentence “At the time of the survey….” Is confusing. Delete the phrase after 292 residents.
Page 3, line 217. Replace the word “hosts”. This makes no sense. Also replace the word compliance, which is poor English.
Page 3, line 219. Create an appendix, which can be available online, which includes all of the letters, scripts, questionnaires, etc. in the original Italian.
Page 4, line 397. End the sentence at “index”. Start a new sentence with “No check…”
Page 4, line 400. Move the sentence about the study being done in Italian to earlier on page 3.
Page 5. Rewrite the table legend so that it is more complete. For example, Descriptive analysis of variables in a study of oral health in nursing homes in Florence, Italy in year”. Add an overall N to the table.
Page 6, line 543. See my comment about the use of the word risk earlier.
Page 6, Improve the legend on Table 2 as recommended for Table 1. Add an overall N
Page 7, Improve the table legend. Add an overall N
Page 7, line 593, see my earlier comment about the use of the word “risk”
Page 8, line 743 and 744. “were accepted” and “structures” are poor English. Please substitute better wording.
Page 8, line 749. “tutors” is not good English. Please substitute a better word.
Page 9, line 822. Fix the spelling of “breeding”
References. The style of abbreviations of journal names is inconsistent. Also remove the square brackets for titles that are not in English. This is picked up from PubMed but is not appropriate in the reference list in a paper.
Author Response
Page 1, line 37. Substitute dental examination for dental visit.
We have corrected it.
Page 1, line 40. Rewrite the sentence beginning “The risk of malnutrition..” to read “The malnutrition score was unrelated to ….” An actual odds ratio was not actually calculated. Add a statistic to this sentence.
We have changed the sentence and added the statistic as requested.
Page 3, line 205. Rewrite this paragraph to read “The objective of this research was….”
We have changed it.
Page 3, line 211. Add, Florence, Italy
We have added it.
Page 3, line 215. The sentence “At the time of the survey….” Is confusing. Delete the phrase after 292 residents.
We have changed the sentence.
Page 3, line 217. Replace the word “hosts”. This makes no sense. Also replace the word compliance, which is poor English.
We have changed the word "hosts" with "residents" and the word "compliance" with "adherence".
Page 3, line 219. Create an appendix, which can be available online, which includes all of the letters, scripts, questionnaires, etc. in the original Italian.
We have created an appendix with the original Italian documents.
Page 4, line 397. End the sentence at “index”. Start a new sentence with “No check…” Page 4, line 400. Move the sentence about the study being done in Italian to earlier on page 3.
We have changed it.
Page 5. Rewrite the table legend so that it is more complete. For example, Descriptive analysis of variables in a study of oral health in nursing homes in Florence, Italy in year”. Add an overall N to the table.
Page 6, Improve the legend on Table 2 as recommended for Table 1. Add an overall N
Page 7, Improve the table legend. Add an overall N
We have rewritten the table legends and added the overall N to the tables.
Page 6, line 543. See my comment about the use of the word risk earlier.
Page 7, line 593, see my earlier comment about the use of the word “risk”
We have changed the expression "risk of malnutrition" with the more proper expression " malnutrition score".
Page 8, line 743 and 744. “were accepted” and “structures” are poor English. Please substitute better wording.
We have changed the expression "were accepted" with the expression "agreed to participate" and the word "structures" with "institutions".
Page 8, line 749. “tutors” is not good English. Please substitute a better word.
We have changed the word "tutors" with the more proper word expression "legal representatives".
Page 9, line 822. Fix the spelling of “breeding”
We have changed with "feeding".
References. The style of abbreviations of journal names is inconsistent. Also remove the square brackets for titles that are not in English. This is picked up from PubMed but is not appropriate in the reference list in a paper.
We have written the references according to the instructions given by the editor, in the section "Instruction for authors".